# Evaluating the implementation of the Reproductive Life Plan in disadvantaged communities: A mixed-methods study using the i-PARIHS framework

Jenny Niemeyer Hultstrand[1¤]*, Ellinor Engström[1], Mats Målqvist[1], Tanja Tydén[1], Nokuthula Maseko[2], Maria Jonsson[1]

1 Department of Women's and Children's Health, Uppsala University, Uppsala, Sweden, 2 Siphilile Maternal and Child Health, Matsapha, Eswatini

¤ Current address: Department of Women's and Children's Health, Akademiska Sjukhuset, Uppsala, Sweden

* jenny.hultstrand@kbh.uu.se

## Abstract

### Introduction

The Reproductive Life Plan (RLP) is a clinical tool to help clients find strategies to achieve their reproductive goals. Despite much research on the RLP from high-income countries, it has never been studied in low- or middle income countries. Together with health workers called Mentor Mothers (MMs), we used a context-adapted RLP in disadvantaged areas in Eswatini. Our aim was to evaluate the implementation of the RLP in this setting.

### Methodology

MMs participated in focus group discussions (FGDs, n = 3 MMs n = 29) in January 2018 and at follow-up in May 2018 (n = 4, MMs n = 24). FGDs covered challenges in using the RLP, how to adapt it, and later experiences from using it. We used a deductive qualitative thematic analysis with the integrated Promoting Action on Research Implementation in Health Services (i-PARIHS) framework, creating themes guided by its four constructs: facilitation, innovation, recipients and context. The MMs also answered a questionnaire to assess the implementation process inspired by normalization process theory.

### Results

The RLP intervention was feasible and acceptable among MMs and fit well with existing practices. The RLP questions were perceived as advantageous since they opened up discussions with clients and enabled reflection. All except one MM (n = 23) agreed or strongly agreed that they valued the effect the RLP has had on their work. Using the RLP, the MMs observed progress in pregnancy planning among their clients and thought it improved the quality of contraceptive counselling. The clients' ability to form and achieve their

**Data Availability Statement:** The data set used in this study contains potentially identifying and

sensitive information about the study participants. There are a limited number of Mentor Mothers working within the organization and, therefore, quotes could be linked to specific persons. Further, during the focus group discussions, personal information (such as age, number of children and relationship status) was shared and sensitive topics such as own experience of family planning and intimate partner violence were discussed. In the Informed consent form, it was stated that 'it will not be possible to link what you have said or written to you personally. No unauthorized persons will take part of any of your answers.' Sharing the full data set is therefore not legally possible. Finally, the National Health Research Ethics Guidelines 2nd Edition, the National Health Research Review Board (NHRRB) in Eswatini states that 'Data generated from research subjects residing in Swaziland shall be the property of the country.' Further, 'Researchers interested in using existing raw data shall seek approval of the NHRRB based on presentation of permission to use the data in question from the original owners of the study.'. This means the research group is not allowed to share data without permission from the NHRRB. Therefore, only a sample of de-identified data will be available upon request. Researchers who meet the criteria to access confidential information, interested in using existing raw data, can access the minimal data set underlying the study by first sending a request to corresponding author, to be granted an official permission. The researchers requesting the data will then need to seek approval of the NHRRB based on presentation of the permission. The contact information provided to NHRRB Office is: 1st floor Christian University Building, Mbhilibhi, Street, Mbabane. Telephone: +268 2404 4905/0865. The authors confirm that if future researchers are unable to contact the corresponding author, they can still request access to the data through the NHRRB. Additionally, a second author's contact information for data access queries is: Maria Jonsson maria. jonsson@kbh.uu.se Cellphone number +46 70 092 62 60

**Funding:** This study was performed with financial support from: 1. Uppsala County Council. Received by TT, grant number AS 2014-0831. https://www. medfarm.uu.se/vetenskapsomradet/namnder-och-kommitteer/alf/ 2. The Family Planning Fund, Uppsala University. Received by JNH in 2018 and 2019, grant number not used. http://www. familjeplaneringsfonden.c.se/ The funders did not play any role in the study design, data collection and analysis, decision to publish, or preparation of the manuscript.

reproductive goals was hampered by contextual factors such as intimate partner violence and women's limited reproductive health and rights.

## Discussion

The RLP was easily implemented in these disadvantaged communities and the MMs were key persons in this intervention. The RLP should be further evaluated among clients and suitable approaches to include partners are required.

## Introduction

A Reproductive Life Plan (RLP) reflects an individual's pregnancy intentions in the context of their personal values and life goals [1–3]. In 2006, Merry K. Moos suggested a set of questions that healthcare providers could ask individuals or couples to make them reflect on their pregnancy intentions and create a plan (Box 1) [3, 4]. This tool is hereafter referred to as the RLP. The RLP aims to empower clients in finding strategies to achieve their reproductive goals, e.g. by providing advice on contraceptives until a pregnancy is wanted and on how to improve pre-conception health [1, 3–6]. Reproductive life plan assessment is recommended in the U.S. as an important part of preconception care [1, 6, 7]. The long-term goal is to decrease the number of unplanned pregnancies and to improve pregnancy outcome [2, 8].

Unplanned pregnancies are associated with adverse maternal and neonatal health outcomes including unsafe abortion procedures, inadequate prenatal care and postpartum depression [9–12]. Some studies also indicate an increased risk for premature birth and low birth weight babies [12–14]. The consequences of unplanned pregnancies are greater in low- and middle income countries and reducing them is a key objective in global public health.

The Kingdom of Eswatini, formerly Swaziland, is a lower-middle income country in Sub-Saharan Africa with an extraordinary high incidence of unplanned pregnancies [10, 15]. The maternal mortality is high (389 per 100'000 live births), teenage pregnancies are common and abortion laws are restrictive [16]. The HIV-prevalence is among the highest in the world at

---

### Box 1. The Reproductive Life Plan tool developed by Merry K. Moos (2006) [4].

1. Desire to have children. *Do you plan to have any (more) children*?

2. Number of children desired. *If so, how many children do you hope to have*?

3. Spacing of children. *How much space do you plan to have between your future pregnancies*?

4. Timing of children. *How long do you plan to wait until you become pregnant*?

5. Plan. *What do you plan to do to avoid pregnancy until you are ready to become pregnant*? *What can I do to help you achieve your plan*?

**Competing interests:** I have read the journal's policy and the authors of this manuscript have the following competing interests: Nokuthula Maseko (NM) is the current and Mats Målqvist was the former Executive Director of Siphilile Maternal and Child Health. This does not alter our adherence to PLOS ONE policies on sharing data and materials.

**Abbreviations:** CDC, Centers for Disease Control and Prevention; FGD, Focus Group Discussion; FP, Family planning; i-PARIHS, integrated Promoting Action on Research Implementation in Health Services framework; MM, Mentor Mother; RLP, Reproductive Life Plan; Siphilile, Siphilile Maternal and Child Health; TICD checklist, Checklist of Determinants of Practice.

32.5% among women [17]. Eswatini lack updated national health data, but in a bordering province in South Africa sharing socioeconomic characteristics with Eswatini, around 71% of all women are overweight, 29% suffer from anemia and 48% from hypertension [18]. These numbers illustrate a country with poor preconception health status and a failure to meet women's reproductive health needs, which motivates interventions for improvement.

Reproductive life planning has been evaluated in high-income settings and is associated with provision of preconception care, increased preconception health knowledge and fertility awareness, but not with contraceptive use [19–23]. Nurse-midwives consider the RLP a health-promoting tool and suggest it should be expanded to other health care professions and arenas [24, 25]. To the best of our knowledge, the RLP has never been tested in any low- or middle income country. The healthcare providers using the RLP have generally been nurse-midwives or general practitioners [19, 21–26], and the use of it among community health workers has not yet been studied. Further, research on the implementation of reproductive life planning is lacking [2, 3, 19]. Implementing effective reproductive health tools used by community health workers could have a significant impact in settings where resources are limited and unplanned pregnancies are common. Therefore, we developed and implemented a context-adapted RLP intervention in a community health setting in Eswatini. The aim of this study was to evaluate the implementation of the RLP among community health workers, called Mentor Mothers (MMs) in this low-income setting.

## Methods

This is an interventional, mixed-method study. The study procedures are presented in Fig 1. To collect qualitative data, we used focus group discussions (FGDs) at baseline in January 2018 (FGD 1) and at follow-up in May 2018 (FGD 2). Quantitative data were collected using one questionnaire at baseline (Q1) and one at follow-up (Q2).

### Setting and study population

Siphilile Maternal and Child Health (Siphilile) is a non-governmental organization that was started in Eswatini in 2012 [27]. Siphilile builds on the Philani Mentor Mother Model developed in the townships of Cape Town, South Africa by Ingrid Le Roux [27]. This model is proven effective on several health outcomes including consistent condom use and prolonged exclusive breastfeeding [28, 29]. Siphilile recruits and educates women from the neighborhoods to become MMs, i.e. peer supporters for women in their living area. MMs share characteristics with community health workers but address multiple health behaviors and are recruited because they are positive deviants, i.e. mothers who despite poor conditions have succeeded in raising healthy children [27, 28]. MMs are trained about basic health issues such

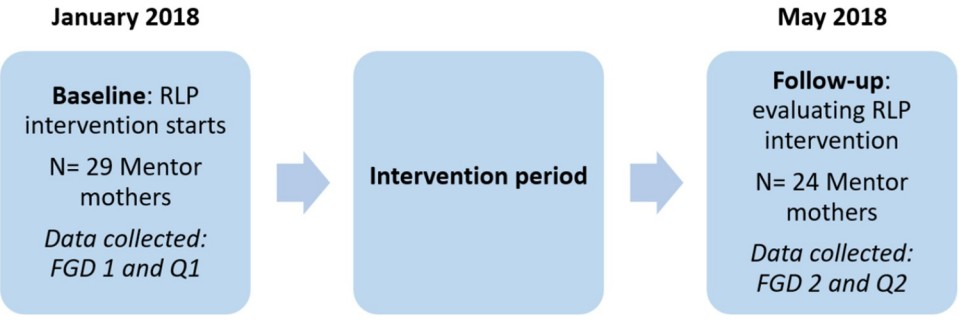

**Fig 1. Study procedures and timeline.**

as HIV-prevention, nutrition and family planning (FP). They support mothers, mothers-to-be and children in their homes through health discussions, basic physical examinations and referrals to hospitals.

In early 2018, Siphilile was active in two low-income settings: The Manzini and Lubombo regions. Manzini is a large city and Matsapha is an industrial hub in the Manzini region. Lubombo is a rural part of Eswatini, where an office was started in 2016. Siphilile's clients constitute a disadvantaged group as most women are unemployed, lack of food is common and the HIV-prevalence is 41%, which is higher than the national average [15, 17]. Seven out of ten pregnancies are unplanned and teenage pregnancies are common [15].

**Sample and recruitment.**   Fifty-three MMs were employed by Siphilile and 29 were asked to participate face-to-face by the first author (JNH), as this number could include several MMs from each geographical region where Siphilile is active. We estimated three to four focus groups at baseline and at follow-up would be sufficient. We allowed for a 25% drop out of informants and scheduled nine to ten informants in each FGD. However, all MMs agreed to participate; 20 MMs were working in the Manzini region and nine MMs in the Lubombo region.

## Study procedures

**The intervention.**   We started the intervention in January 2018 and a tailored RLP tool was developed together with the Siphilile management and the MMs during two work-shops with JNH (Fig 1). The MMs gave suggestions on what topics to discuss when clients wanted or did not want children, and JNH suggested additional topics based on current research and recommendations on preconception care and on reproductive life planning [1, 5]. The adapted RLP tool was reviewed by the management and by the research team. Tailoring the RLP was essential, as cultural appropriateness and contextual relevance is important for successful implementation [30]. The MMs and their supervisors were trained in using the RLP by JNH. The work-shop was followed by learning activities in preconception health and contraceptive counselling. The MMs were instructed to use the RLP with their clients and were given the adapted RLP tool (S1 Appendix) and a report sheet to document which clients that had been approached. The MMs were encouraged to include partners in the discussion if the client approved.

**Data collection.**   After the RLP training, FGDs with the MMs were held. All FGDs were moderated by JNH in English as all MMs speak English fluently. We used a semi-structured discussion guide (S2 Appendix). Questions were open-ended, covering perceptions of using the RLP, factors affecting FP discussions and partner involvement. The discussion guide was checked by the research team for cultural accuracy and relevance before the FGDs started. The MMs also answered a questionnaire (Q1), covering age, years working as a MM, relationship status, number of children, as well as own FP history.

We evaluated the intervention in May 2018. Four new FGDs were held with the MMs, with partly new group combinations. The FGDs covered experiences from using the RLP and determinants of its use. The semi-structured discussion guide included open-ended questions as well as questions on the implementation process inspired by the Checklist for Determinants of Practice (TICD checklist) developed by Flottorp et al. [30, 31]. Determinants of practice are factors that can enable or disable implementation of an innovation and the checklist is used to identify those. Determinants are categorized into guideline factors, individual health professional factors, patient factors, professional interactions, incentives and resources, capacity for organizational change as well as social, political, and legal factors [30, 31].

The MMs answered a questionnaire (Q2) covering experiences from and implementation of the intervention. Q2 was pilot-tested in a group of four MMs who gave their feedback and amendments were made where needed. Questions were adapted from the NoMAD instrument [32, 33]. NoMAD is an implementation measure based on a social theory to understand the dynamics of implementing complex health interventions, called Normalization Process Theory [33]. Key constructs of this theory include *coherence* (sense-making), *cognitive participation* (engagement to sustain a practice), *collective action* (work done to enable the intervention to occur) and *reflective monitoring* (evaluation of the intervention's benefits and costs) [33]. Using the NoMAD offers a structured assessment of implementation processes from the MMs' perspectives [33].

## Data analysis

**Qualitative data and the use of i-PARIHS.**   Data collection stopped when the research team deemed that data saturation was reached. FGDs were audio-recorded and transcribed verbatim by JNH, starting immediately after each FGD to see if any of the topics needed clarification from the MMs. The transcripts were not returned to the participants for comments or correction. Initial ideas or themes were noted as well as memos, written directly after each FGD finished. Recordings were listened to and transcripts were read multiple times by JNH and the second author (EE). Guided by the research questions, descriptive and in vivo codes were generated openly by JNH and cross-checked by EE. In vivo codes were used as we wanted to prioritize the MMs' voices [34]. No software was used.

After coding, we used a deductive approach driven by the integrated Promoting Action on Research Implementation in Health Services (i-PARIHS) framework [35, 36]. This is a framework to describe implementation processes. Its ancestor, the PARIHS framework was first published in *Implementation Science* in 1998 [35], aiming to help explain and predict why the implementation of evidence into practice is successful or not. The theory that the i-PARIHS framework builds upon claims that successful implementation is a function of four constructs: the *innovation*, the *recipients* and the inner and outer *context*, and *facilitation* is the active component. Each of these constructs have characteristics that either enables or disables implementation. For example, to be successfully implemented the *innovation* should be usable, fit well with existing practices, and give observable results. *Recipients*, i.e. individuals who are affected by and influence the implementation, should have enough knowledge, time and support to perform the innovation. Implementation is also affected by *contextual* factors such as working culture, leadership and policies. *Facilitation* activates implementation through assessing and responding to characteristics of the innovation, its recipients and the context where it is implemented [35]. Both the PARIHS and the i-PARIHS framework have formerly been used in similar settings to evaluate implementation of innovations [37, 38].

Codes were grouped into the i-PARIHS constructs (facilitation, innovation, recipients and context), constantly going back to the transcripts to make sure our interpretations of the codes were valid. The same code could appear in more than one construct and codes that did not fit into any of the constructs were saved for potential later categorization. We used thematic analysis by Braun et al. [39], and mainly applied a realist perspective as we sought to report the experiences and reality of the participants. After grouping the codes, they were printed on paper and we searched for patterns among the codes within each construct. We manually grouped codes that had a pattern capturing something relevant to the research questions and created themes based on these codes. Each theme was controlled to make sure that it was supported by the codes and if it was congruent with the entire data set. The creation of themes was undertaken in groups of two to three members of the research team during five work-

shops, and different constellations of researchers participated in each work-shop. Finally, the themes were reviewed, refined and negotiated within the whole research team. The participants did not provide feedback on the themes.

**Quantitative data.** Quantitative data from Q1 and Q2 were analyzed using frequencies in IBM SPSS Statistics 23.

## Ethical concerns

The National Health Research Review Board in Eswatini (SHR010/2018) and the Regional Ethical Review Board in Uppsala (2017–514) approved the study. The MMs were requested face-to-face by JNH to participate in the study. They were provided with oral and written information, stating that they could withdraw at any time without needing to state why and without consequences. Voluntary participation was emphasized repeatedly by JNH and by Siphilile's Executive Director. All the MMs signed an informed consent.

The FGDs were held at Siphilile's office in Matsapha and at a health facility in the Lubombo region, in rooms separated from other staff. This location was deemed appropriate as the MMs were participating in their role as health professionals. Only the MMs and JNH were present during the FGDs. Pseudonyms were used for the MMs. The full transcripts were only available to the research team except for NM who is the Executive Director of Siphilile.

The MMs received no financial compensation except for lunch and money for transportation.

## Results

All MMs participated at baseline (n = 29) and 24 at follow-up. The reasons for the five MMs' drop-out were not specifically requested due to ethical reasons, but reasons mentioned were sick-leave, loss of relative, and health care appointments. There were between four and ten participants in each FGD. FGDs at baseline lasted between 37 to 45 minutes (mean 41 minutes) and at follow-up between 52 to 95 minutes (mean 66 minutes). Table 1 presents characteristics

**Table 1. Characteristics of the study population, 29 Mentor Mothers.**

| Variable | Frequency n (%) |
|---|---|
| Relationship status | 29 (100) |
| *Married* | 16 (55) |
| *Partner* | 4 (14) |
| *Single* | 9 (31) |
| Discussed number of children with partner | 28 (97) |
| *Yes* | 13 (45) |
| *No* | 15 (52) |
| Have had an unplanned pregnancy | 29 (100) |
| *Yes* | 20 (69) |
| *No* | 9 (31) |
| Have discussed contraception with partner | 28 (97) |
| *Yes* | 20 (69) |
| *No* | 8 (28) |
| Current use of contraception | 29 (100) |
| *Yes* | 14 (48) |
| *No* | 15 (52) |

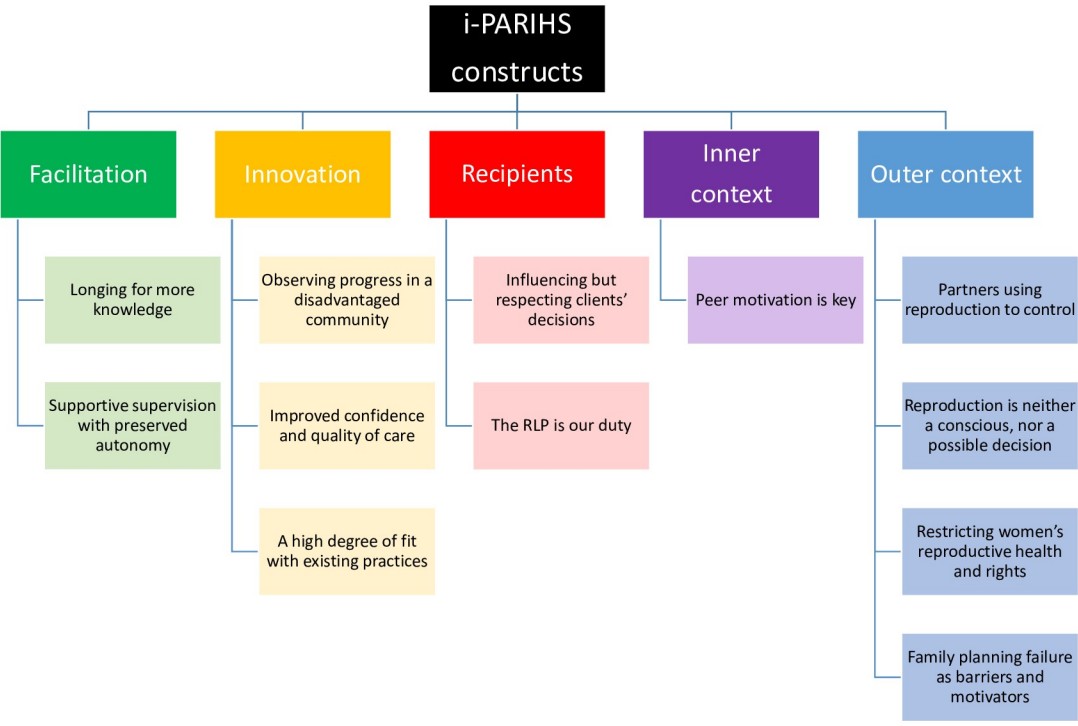

**Fig 2. Themes identified in each construct of the i-PARIHS framework.**

of the MMs, who had a median age of 36 years (range 26–59 years) and median number of children was three.

Qualitative as well as quantitative findings from Q2 are presented below sub-headings representing each of the constructs in the i-PARIHS framework (Fig 2). The complete findings from Q2 are presented in Table 2. Quotes from FGDs at baseline are labelled 'FGD 1' and quotes from FGDs at follow-up 'FGD 2', followed by the group number.

## Facilitation

**Theme: Longing for more knowledge.** The MMs expressed that the RLP training had been helpful. One discussed how knowledge about the interaction between HIV-medication and progesterone had been crucial for her FP counselling. This interaction has been described by Nanda et al. [40]. The MMs also expressed how they needed to be equipped with more knowledge on contraception to be able to fight FP myths and to address contraceptive failure:

'I think we still need more about the contraceptives, and straight to the contraceptives, how do they work exactly? Yes, because the clients they do have experiences with these things. [. . .], say 'I have taken that [contraception], but I was pregnant'. So how to address that?'

Qondile FGD 2:2

During the FGDs, the MMs suggested provision of additional training through seminars or open group discussions as well as learning material such as handouts and brochures. This was provided at follow-up. JNH also started a WhatsApp group with the MMs and their supervisors with the aim of providing weekly updates on contraception.

**Table 2. Mentor Mothers' (n = 24) opinions on using the Reproductive Life Plan in Eswatini.**

| Question | Strongly agree n (%) | Agree n (%) | Neither agree nor disagree n (%) | Disagree n (%) | Strongly disagree n (%) | Missing n (%) |
|---|---|---|---|---|---|---|
| I can see how the RLP is different from our usual ways of working | 2 (8) | 12 (50) | 2 (8) | 5 (21) | 3 (13) | 0 |
| Staff in this organization (manager, supervisors, MMs) have a shared understanding of the purpose of the RLP | 10 (42) | 8 (33) | 6 (25) | 0 | 0 | 0 |
| I can see the potential value of the RLP for my work | 12 (50) | 11 (46) | 0 | 0 | 1 (4) | 0 |
| I believe that participating in the RLP is a legitimate (valid) part of my role as a MM | 14 (58) | 9 (38) | 0 | 0 | 1 (4) | 0 |
| I will continue to support the RLP | 16 (67) | 7 (29) | 1 (4) | 0 | 0 | 0 |
| I can easily combine the RLP with my existing work | 14 (58) | 9 (38) | 1 (4) | 0 | 0 | 0 |
| I have confidence in other Mentor Mothers' ability to use the RLP | 8 (33) | 10 (42) | 4 (17) | 1 (4) | 0 | 1 (4) |
| Enough training is provided to enable us Mentor Mothers to use the RLP | 11 (46) | 8 (33) | 3 (13) | 2 (8) | 0 | 0 |
| Management provides enough support for the RLP | 9 (38) | 8 (33) | 4 (17) | 3 (12) | 0 | 0 |
| We (Mentor Mothers) agree that the RLP is worthwhile | 12 (50) | 12 (50) | 0 | 0 | 0 | 0 |
| I value the effects that the RLP has had on my work | 14 (58) | 9 (38) | 1 (4) | 0 | 0 | 0 |
| Feedback about the RLP can be used to improve it in the future | 13 (54) | 10 (42) | 1 (4) | 0 | 0 | 0 |
| I can change how I work with the RLP depending on the situation | 8 (33) | 13 (54) | 2 (8) | 1 (4) | 0 | 0 |
| The RLP makes it easier for me to support clients forming a goal about if they want children or not | 17 (71) | 7 (29) | 0 | 0 | 0 | 0 |
| The RLP makes it easier for me to discuss family planning with my clients | 18 (75) | 6 (25) | 0 | 0 | 0 | 0 |
| The RLP makes it easier for me to discuss how to improve health before pregnancy with my clients | 16 (67) | 5 (21) | 3 (12) | 0 | 0 | 0 |
| The RLP makes it easier for me to discuss how to keep the ability to have children with my clients | 12 (50) | 10 (42) | 0 | 0 | 0 | 2 (8) |

Results from Q2 (Table 2) showed that most MMs agreed that enough training was provided (77%, n = 19). Three neither agreed nor disagreed and two disagreed.

**Theme: Supportive supervision with preserved autonomy.** The MMs emphasized that their supervisors supported them in using the RLP by repeatedly asking them if they used it, and by assisting if there were any challenges. Supervision in the RLP, specifically, was integrated in the ordinary monthly supervision. More supervision was seldom requested by the MMs, who generally had a high degree of self-reliance in meeting their clients.

'I think our supervisors, they supervise it, because we use to talk about it even when they are visiting them [the clients]. So, I don't think there is more need of it.'

Gugu FGD 2:2

Most of the MMs (75% n = 18) agreed that staff within Siphilile (manager, supervisors, other MMs) had a shared understanding of the purpose of the RLP and a majority of the MMs (71% n = 17) agreed that the management provided enough support for the RLP. Three MMs disagreed (Table 2).

## Innovation

**Theme: Observing progress in a disadvantaged community.** The innovation (the RLP) was seen as advantageous and results from using it were clearly observable by the MMs in these disadvantaged communities with high rates of unplanned pregnancies. At baseline, the

perceived benefits of the innovation were that unplanned pregnancies would decrease, that it would facilitate discussion with partners and increase clients' knowledge. This foreseen progress would then have an impact on the whole community:

> 'Because you have to begin here, from us, from our home, from our sisters in law. From explaining to the community, it will work. They [the clients] will see the goodness of this. That it will limit, it will save us from poverty. We are poor because we have many unplanned pregnancies, children take much responsibilities in matter of financial'
>
> Temalangeni FGD 1:2

Some of the beliefs mentioned above were confirmed at follow-up, when the MMs expressed that clients had started to plan pregnancies and use contraception. The teenagers were highlighted as a group that had benefitted much from the intervention. However, more resources were needed to reach teenagers that were not pregnant or had not given birth, as they are not part of the Siphilile intervention.

> 'With the RLP-tool, we are able to encourage them [the clients] to stop giving birth to unwanted and or unplanned children.'
>
> Babhekile FGD 2:1

> 'It has had a great impact on our client. I once had a client where by maybe yearly, give birth, yes, [. . .] but now, she's on implant and even her husband saying she, he appreciate'
>
> Thandiwe FGD 2:2

All except one MM (n = 23) agreed or strongly agreed that they valued the effect the RLP has had on their work (Table 2).

**Theme: Improved confidence and quality of care.** The MMs expressed how the RLP intervention had improved their confidence in contraceptive and preconception counselling. The RLP-questions were perceived as advantageous as they opened up discussions with the client and enabled reflection, both from the client's and the MM's sides:

> 'What I see on these questions is that there are things that are opening her mind and even my mind. When I ask the question [Do you want to have any more children?], I listen to the answer and then I can tell her "you can do this". Sometimes I don't know what to say, but now I know.'
>
> Nosipho FGD 2:3

> 'I think it has been an eye-opener to me and to some of my clients. [. . ..] Like, changing your [preconception] lifestyle it has been one of the most important things I have experienced, and I have learned to, shared with my clients.'
>
> Babhekile FGD 2:1

Before the RLP intervention, the MMs had discussed FP but focused solely on contraception and not on preconception health. Former counselling had been limited to postpartum women before the six-week control at the clinic, briefly mentioning the contraceptive methods available, and follow-up was not provided. With the RLP, the MMs perceived the

contraception care they provided as improved, since the RLP enabled them to elaborate more on contraceptive methods and to follow-up on their clients:

'I have to make a good follow-up, because last time [before the RLP] I was talking that 'you must do family planning, you must do family planning', now I have to elaborate'

Nomzamo FGD 2:3

All the MMs (n = 24) agreed that the RLP made it easier for them to support clients in forming reproductive goals, as well as in discussing family planning and how to improve pre-conception health (Table 2).

**Theme: A high degree of fit with Siphilile's practices.**   Implementing the RLP in daily practice was easy as it fit well with existing practices. Many parts of the intervention were not new, as contraception, HIV-testing and birth spacing were already within the MMs' curriculum. The RLP intervention was perceived as best when used in one-on-one discussions and when the MMs listened carefully to the client; practices that were already standard. At follow-up, the MMs expressed how the RLP was easy to integrate, required little effort and how it fitted well in their daily schedule:

'[RLP] It's like, it's a daily thing, like you are moving from house to house, the work that we are doing. [. . .]. So, it's not hard. It's a daily thing that you are doing in the field, it's simple.'

Nosipho FGD 2:3

On the statement 'I can see how the RLP is different from our usual ways of working' (Q2), eight MMs disagreed and twelve agreed. All except one agreed that they could easily combine the RLP with their existing work (Table 2).

## Recipients

**Theme: Influencing but respecting client's decision.**   When the MMs were asked about how they applied the RLP, some expressed they asked rhetorical questions, such as 'do you want to go back to the labor ward?' instead of the instructed patient-centered question 'do you want to have any more children?' Some MMs were concerned when clients' reproductive intentions were not in line with the well-being of the whole family, as the MMs were well aware of the consequences of unplanned childbearing. This family-centered approach sometimes created a dilemma when trying to help women achieve their reproductive goals:

'I have a client that has had her fifth child. When I ask her "do you want another child?", and she says, "Yes soon", I will be like "Oh, no!" Because I know the importance of birth spacing, I will be like no. . . Then she can get angry.'

Phetsile FGD 1:1

Some expressed how they had tried to influence the client's decision by providing information about dangers associated with childbearing and the benefits of birth spacing. Nevertheless, they emphasized the importance of not trying to push the client, and of always respecting her decision:

'[Using the RLP] we are able to make them talk. But you can say it now and again, but it's them to take a decision. You have given them the options but it's them to take their own decisions'

Thandiwe FGD 2:2

**Theme: The RLP is our duty.**   Most of the MMs were strongly motivated to implement the RLP. They expressed the importance of providing correct information to the client, of teaching her until she understands, and of discussing both advantages and disadvantages with different contraceptive methods. This, they believed, would help create an alliance with their clients and lack of knowledge would make clients doubt them. The goal, or rather the responsibility, of the MMs was to help clients make informed decisions.

'The RLP it will reduce most of the poverty that we see now in the community. Yeah, because having a lot of children, unplanned children, there are so many, then if you don't introduce that, you are actually also bringing poverty yourself because you've got the information, but you are keeping it.'

Nosipho FGD 2:3

All except one MM (n = 23) agreed that participating in the RLP was a legitimate part of their role and that they will continue to support the RLP (Table 2).

## Inner context

**Theme: Peer motivation is key.**   The MMs expressed how their role as peer supporters was key in successful RLP-counselling. The relationship between the MM and the client, starting from when she first became pregnant, and the client's trust in their MM, facilitated and deepened the RLP discussions:

'I think this thing [the RLP] is going to work. Because we have already bonded with our clients. [. . .] Now it will be superb, because we are talking to somebody who knows me, and we've been together for the past 4 years, it will help them to think outside the box.'

Tengetile FGD 1:1

The MMs suggested that the use of peer motivation in pregnancy planning could be expanded to include teenage clubs and training of Mentor Fathers. Finally, the MMs took pride in their role as peer supporters and hoped their own experiences would help their clients:

'You know what, I know what I want because of the mistakes I have made. I've take up a stand because of the mistake I've had. I have had maybe unwanted pregnancies, maybe all of my kids are unwanted. I've never planned kids. I think, those who plan they are rich. [. . .] Maybe we can say, please learn from my mistake—I have had three unwanted pregnancies, you know.'

Temalangeni FGD 1:2

## Outer context

**Theme: Partners using reproduction to control.** The MMs explained that reproduction was sometimes used by partners as a way of exercising power on one another. While men were sometimes making women pregnant as a way of controlling them, women also tried to exercise power through reproduction, by 'tricking' men to make them pregnant. Thereby they hoped for love or financial support for them and their child.

'The girlfriend loves the boy and she feels like "I will catch him by giving him the baby". So, they just neglect themselves, the girls. Even the boy, sometimes feel that if the woman can have more babies she cannot look for other men because she will be busy with the children all the time.'

Nolwazi FGD 1:2

'She gets the income by giving birth. She'll follow the men, sometimes she follow the men, he gives her nothing, but keep on giving birth.'

Nomzamo FGD 2:3

**Theme: Reproduction is often neither a conscious, nor possible decision.** The MMs expressed that on the individual and family level, many women and couples had not reflected upon the wanted number of children. In some cases, this was attributed to vulnerability such as depression and poverty. Other MMs expressed that many women *know* how many children they want, and they repeatedly expressed that most children were unplanned or unwanted and that almost all their clients wanted to prevent pregnancies.

'I don't know if it is our culture or what, sometimes it is very difficult that you find couples discussing of their way forward. Maybe now with the new generation, but to be honest, from my generation with the experience I have had, we had never sat down and discussed about how many children really are we planning to have as a family, and that's a fact. Yeah, so the children you give birth to, they are all unplanned.'

Nok´lunga FGD 1:1

Discussing number of children with partners outside a marriage was considered a taboo according to Swati culture and disagreement between partners was common, men generally desired more children than women. Men were often decision-makers on both contraception and on sexual intercourse. Yet, women were repeatedly blamed when they became pregnant as well as if the couple could not conceive. The MMs gave testimony about abusive husbands, and encouraged by the MMs, some clients hid their contraception from their partners, as revealing would result in partner refusal of dual protection with condom:

'So, I say, "don't tell your husband". You see that is not good, that you hide something [contraception] from your partner, but I've said so, "don't tell it, don't tell him", so that he will put the condom.'

Phangisile FGD 2:2

'By that time she was having four children. Then she decided, "I want to go with you, can you take me to the clinic?" I said, "Hey what are you going to do if the man hears that

you've been to the clinic?" She says "no, I won't tell him, it will be my responsibility". We went here at the Nazareen [hospital]. They put her on implant. [. . .] One day she told me, "Sawubona, he has find out". [. . .] Okay, after that there was a big quarrel, a fight also.'

Tandzile FGD 1:2

The MMs also expressed that men had limited power in reproductive decisions as they lack knowledge and are not involved in FP. This hampered their ability to provide emotional support. The importance of the partner's support and involving him in the RLP conversations was highlighted in all FGDs as key for successful family planning. However, due to the partner being absent during daytime, this was seldom possible:

'The problem is the husband is always at work.'

Melokhule FGD 2:4

'They [the men] don't know if you have done it [FP], they are not involved. [. . .] maybe they lack knowledge, they need to be supportive, also they need to be part of this thing. They are the ones who will suffer the consequences of having this unplanned pregnancies.'

Temalangeni FGD 1:2

**Theme: Restricting women's reproductive health and rights.** Women's reproductive health and rights were also restricted on a societal and cultural level. Achieving reproductive goals was hindered on a national level by legislations such as women needing a family member's signature to get access to sterilization. Concerns were raised about the quality of healthcare and attitudes among family planning nurses. Nurses not informing about all contraceptive methods available, only providing brief and sometimes incorrect information, were commonly expressed by the MMs:

'They just chase you away and say you cannot have FP because you have these [varicose] veins and high BP [blood pressure].'

Temalangeni FGD 1:2

Other structural barriers included few contraceptive methods available at government clinics, preconception care limited to HIV-testing, and lack of money for contraception and healthy food.

'You have to go for Manzini for a loop. Plus, it's costly.'

Gcinile FGD 2:4

Finally, important concerns were raised about stigmatizing teenagers that wanted to use contraception, as the clinics are not welcoming women below the age of 18.

**Theme: Family planning failure as barriers and motivators.** In this context, the MMs described problems with side-effects and a high incidence of contraceptive failure and of unplanned pregnancies. This carried great consequences for both the woman, the child and the family; school drop-out for young mothers, financial constraints in the family and not being taken care of as a child. These consequences of family planning failure was a rationale

for engaging in the RLP and for teenagers, going back to school was a strong motivating factor.

> 'You see the reason and the need to preach about these things [Pregnancy planning and preconception health]'

> Babhekile FGD 2:1

Although the MMs own and clients' experiences of FP failure often motivated the MMs, it sometimes created doubts. The MMs expressed that using the RLP was challenging with women that had experienced contraceptive failure, which was common among HIV-positive women. This made it hard for the MMs to motivate those clients.

> 'As for me, not as a Mentor Mother, I felt it has not worked for me. [. . .] Which method should I choose? Because I don't want ten children, but I'm still confused, what must I do?'

> Noluntu FGD 1:2

## Discussion

Using the RLP was feasible and acceptable among the MMs, i.e. the health workers, in these resource-limited communities. We identified key factors for the implementation of the RLP using the i-PARIHS framework. One key factor was that the *innovation*, the RLP, fit well with existing practices. Further, supportive supervision *facilitated* its implementation. The MMs, the *recipients*, were engaged in using the RLP and valued the effect it has had on their work. Using the RLP, the MMs observed progress in pregnancy planning in their communities and thought it had improved the quality of preconception and contraception care. It was evident that the formation and achievement of reproductive goals was much affected by factors in the outer *context* such as women's limited reproductive health and rights.

The particular role of a MM was highlighted in the themes as key in successful RLP counseling. The established relationship between the MM and the client was advantageous, as trusting relationships between clients and providers are important, especially when discussing pregnancy and reproduction [24, 41]. Recognizing the lack of, and need for, reproductive planning is necessary to initiate a conversation or intervention [8]. As the MMs are peers from the same context, it was easy for them to do this, as well as helping clients managing barriers for reproductive life planning. Most of the MMs had experienced unplanned pregnancies themselves and understood that the concept of pregnancy planning does not feel attainable to all women [3, 41–43], especially for women with low income [42, 44]. Peer support thus offers significant advantages in RLP discussions in this context. Future studies need to assess the use of RLP in low-income settings among of community health workers that are not peer supporters.

The hypothesized consequences of forming an RLP is a clear pregnancy intention, which may lead to contraception care or preconception care [2]. Few clients wanted more children and consequently, preconception care was seldom mentioned in the FGDs, since the MMs' then had focused on contraception care. In contrast to former studies [20, 23], the MMs reported that using the RLP led to increased contraceptive use among their clients. This finding was supported by the report sheets in which the MMs noted contraceptive use after RLP counselling. Future studies should compare contraceptive use among clients receiving RLP care compared to standard care or other counselling tools, to see if the RLP is an effective way to increase contraception use.

Several screening tools for pregnancy intentions similar to the RLP are available, such as 'One Key Question', 'Family Planning Quotient' and 'Pregnancy, Attitudes, Timing and How important is pregnancy prevention' [42, 45–47]. In this intervention, the MMs had adapted the RLP questions and used other phrasings to screen for pregnancy intentions. The MMs also applied a family-centered approach when using the RLP as they focused more on the well-being of the whole family than on the individual's reproductive goals, as is done in traditional RLP counselling [2, 3, 8]. Being able to modify how to work with an intervention is important for its implementation [33]. In a study comparing One Key Question with Family Planning Quotient, both clients and providers rated the instruments equally [48]. Hence, the choice of instrument may not be very important as long as the provider is tactful and nonjudgmental, which is essential [24, 25, 42]. Therefore, interventions should focus on how healthcare providers best counsel individuals and families with openness and respect.

In this setting, factors in the outer context often hindered women from forming reproductive goals and from achieving them. We found that non-violent power and control tactics from partners on women's reproductive health were common, such as partner's refusal of condom and women needing to hide contraception. These are examples of reproductive coercion or reproductive control and is a type of intimate partner violence [49]. Poor partner support and reproductive control has been reported from similar settings [50–52], and sexual violence is extremely common in Eswatini [53–55]. Interestingly, we also found that women exercised power on men using pregnancies as a means to receive financial and emotional support. Lack of reproductive control among men as well as women have formerly been reported in Uganda, emphasizing that these phenomena are multi-dimensional and both men and women assert reproductive agency at different times and to different extents [52].

Gender and power dynamics as well as absent partners made it hard for the MMs to include partners in the RLP. Reproductive life planning is supposed to be inclusive of both sexes [8], and the importance of including partners was highlighted in all the FGDs. Partner's disapproval on contraception and partnership miscommunication is a primary barrier for contraceptive use in Sub Saharan Africa [56, 57]. Teenagers were also highlighted as a group in need of more support as non-pregnant teenager are not part of the Siphilile intervention. This was not surprising as teenagers in this setting are less likely to use contraception and have a more than two-folded increased odds of unplanned pregnancy compared to older women [15]. The MMs suggested the use of teenage clubs as well as training of Mentor Fathers could help increase teenagers' and men's knowledge and challenge cultural norms and stigma around contraception. Future studies have to assess the value of these approaches in this setting.

Using the i-PARIHS framework offered a structured reporting of implementation of the RLP and our themes fitted well into its constructs. Former studies using the i-PARIHS have discussed context on a local, organizational and external health system level [35, 37, 58]. Similarly, the TICD checklist acknowledges context by including social, political and legal factors [30, 31]. Neither i-PARIHS nor TICD addresses culture, social system or hegemony as part of the context. In this setting, the greatest barriers for implementing the RLP were related to contextual factors such as vulnerability among clients, cultural barriers for partner discussions on reproduction and intimate partner violence. We believe acknowledging context as a broader concept in the i-PARIHS-framework and in the TICD checklist could benefit future implementation research in both low- and high income settings.

The constructs in the i-PARIHS often overlap and the inter-play between them is generally poorly understood [35]. In this study, this frequent overlap was illustrated by this quote: 'You see the reason and the need to preach about these things [Pregnancy planning and

preconception health]' Babhekile FGD2.1. This quote describes the disadvantaged *context* as a source of *motivation* for the recipients (the MMs), as well as an advantage of the *innovation* (the RLP).

## Methodological considerations

To our knowledge, this is the first study on implementation of the RLP in a low- or middle income country and the first in which the RLP is used by community health workers or MMs. The FGDs provided rich data and we performed a robust thematic analysis guided by a comprehensive framework. All authors, with diverse backgrounds, actively participated in the analysis and providing both insiders' and outsiders' perspectives add trustworthiness to the study. This study contributes both to reproductive health science, especially on RLP, and to implementation science.

Jenny Niemeyer Hultstrand (JNH) is a female Swedish medical doctor and PhD-student. In 2016, JNH spent eight weeks working with Siphilile on a study about unplanned pregnancies [15]. During this period, she joined the MMs in their fieldwork, observing their counselling of clients. The MMs knew that the rationale for the study was the high incidence of unplanned pregnancies [15], and that JNH's personal goal was to help improve family planning. This was her first qualitative research project. Mats Målqvist is a male Swedish medical doctor, professor in global health and he was a permanent resident in Eswatini for three years. He was the former executive director of Siphilile and was in charge of recruiting the MMs in Matsapha and knew all of them well. Nokuthula Maseko is a female Swazi registered nurse midwife. She is the current Executive Director of Siphilile and has many years' experience of working with sexual and reproductive health in Eswatini.

A limitation of this study is that both the RLP training and moderating of the FGDs were performed by JNH. This may have created a social desirability bias and potentially skewed the results in a positive direction. To minimize this risk we added anonymous questionnaires (Q2). Since results from the FGDs were supported by results from Q2, this triangulation of data validated our findings.

We did not systematically collect data on number of clients approached, which could have been valuable when interpreting the results. Further, our findings are limited to data from MMs. Evaluating the implementation of the RLP using other informants and means, such as observational studies or interviews with clients, would be valuable. However, we believe our former collaboration with Siphilile and the MMs facilitated implementation and created a trust that enabled open discussions, as sensitive topics were discussed freely during FGDs. This may also explain why all MMs agreed to participate at baseline. Implementing the RLP in other settings without former collaboration may be more challenging but can add important knowledge on other factors affecting its implementation.

## Conclusions

The RLP intervention was feasible and acceptable among the health workers, called Mentor Mothers, and fit well with existing practices. The MMs' role as peer supporters was key in RLP counselling. Using the RLP, the MMs observed progress in pregnancy planning in their communities and thought it improved their quality of contraceptive counseling. Forming and achieving reproductive goals was hampered by contextual factors such as women's limited reproductive health and rights as well as intimate partner violence. The RLP should be implemented and evaluated in similar settings but suitable approaches to include partners as well as teenagers are needed.

## Supporting information

**S1 Appendix. The adapted Reproductive Life Plan tool.**
(PDF)

**S2 Appendix. Focus group discussion guide.**
(PDF)

## Author Contributions

**Conceptualization:** Jenny Niemeyer Hultstrand, Mats Målqvist, Tanja Tydén, Maria Jonsson.

**Formal analysis:** Jenny Niemeyer Hultstrand, Ellinor Engström, Mats Målqvist, Tanja Tydén, Nokuthula Maseko, Maria Jonsson.

**Funding acquisition:** Jenny Niemeyer Hultstrand, Tanja Tydén.

**Investigation:** Jenny Niemeyer Hultstrand.

**Methodology:** Jenny Niemeyer Hultstrand.

**Project administration:** Jenny Niemeyer Hultstrand, Mats Målqvist, Nokuthula Maseko, Maria Jonsson.

**Supervision:** Mats Målqvist, Tanja Tydén, Maria Jonsson.

**Validation:** Ellinor Engström, Mats Målqvist, Nokuthula Maseko.

**Visualization:** Jenny Niemeyer Hultstrand.

**Writing – original draft:** Jenny Niemeyer Hultstrand.

**Writing – review & editing:** Jenny Niemeyer Hultstrand, Ellinor Engström, Mats Målqvist, Tanja Tydén, Nokuthula Maseko, Maria Jonsson.

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
