## [Decision Letter · Decision Letter 0]

6 Apr 2020

PONE-D-19-35394

Evaluating the Implementation of the Reproductive Life Plan in Disadvantaged Communities: a Mixed-Methods Study Using the i-PARIHS Framework

PLOS ONE

Dear Jenny Hulstrand,

Thank you for submitting your manuscript to PLOS ONE. After careful consideration, we feel that it has merit but does not fully meet PLOS ONE’s publication criteria as it currently stands. Therefore, we invite you to submit a revised version of the manuscript that addresses the points raised during the review process.

ACADEMIC EDITOR: Please insert comments here and delete this placeholder text when finished. Be sure to:

Indicate which changes are required versus recommended for acceptanceAddress any conflicts between the reviewsProvide specific feedback from your evaluation of the manuscript

We would appreciate receiving your revised manuscript by 20 May. To enhance the reproducibility of your results, we recommend that if applicable you deposit your laboratory protocols in protocols.io, where a protocol can be assigned its own identifier (DOI) such that it can be cited independently in the future. For instructions see: http://journals.plos.org/plosone/s/submission-guidelines#loc-laboratory-protocols

We look forward to receiving your revised manuscript.

Kind regards,

Sharon Mary Brownie

Academic Editor

PLOS ONE

Journal Requirements:

"I have read the journal's policy and the authors of this manuscript have the following competing interests:

Nokuthula Maseko (NM) is the current and Mats Målqvist was the former Executive Director of Siphilile Maternal and Child Health."

4. Please upload a copy of Figure 3, to which you refer in your text on page25. If the figure is no longer to be included as part of the submission please remove all reference to it within the text.

Additional Editor Comments (if provided):

Reviewers have identified very significant and major deficits within the current manuscript. The nature of comments related to methodology and reporting do mean that a major rewrite is required to bring your paper to standard. For this reason I have allocated six weeks rather than the usual four weeks revision time. It is essential that each and every point is considered and addressed in full. I also recommend that you visit the Equator Website https://www.equator-network.org/ and the relevant reporting guidelines such as COREQ guidelines for reporting qualitative data https://www.equator-network.org/reporting-guidelines/coreq/ Please download and check your work against recommended guidelines and this will help improve the restructuring and content of your manuscript.

Reviewers' comments:

Reviewer's Responses to Questions

**Comments to the Author**

1. Is the manuscript technically sound, and do the data support the conclusions?

The manuscript must describe a technically sound piece of scientific research with data that supports the conclusions. Experiments must have been conducted rigorously, with appropriate controls, replication, and

---

## [Author Response · Author response to Decision Letter 0]

27 May 2020

Please use the separate file: 'Response to reviewers' for an optimized layout and for page references.

REVIEWER 1:

Despite the study might be important in nature especially for the context where it has been conducted. However, the report of the study / the manuscript is poorly written and confusing. The study methods section is lacking major section i.e the study design and does not describe the study settings, participants, data collection procedures clearly. Being a mixed method study also adds to the challenge, as the authors has to describe two methods that has been used, two data analysis procedures, and two results that has been found. They also has to discuss these two results and integrate it in the discussion section. ALL this are not sufficiently done in this manuscript which lead to a more lack of clarity and confusion.

Authors' answer: Thank you for reviewing our manuscript and we are grateful for your feedback. We agree that it is a challenge to use a mixed-methods approach but think it adds validity to our findings. We have tried to acknowledge your criticism point-by-point below and we hope you find our responses satisfying. Please see figure 1 to follow the data collection procedures.

Below are some specific comments: 

The introduction is not adequate. It does not set the scene adequately for the reader to understand the problem or realize its importance. Needs to explain more about the importance of family planning, consequences of poor planning, especially in a low- and middle-income countries/areas and in community settings. More contextual data about the study setting is also necessary, no information at all presented to justify the need for this study or highlighting the significance of this problem to the same context. Also the general organization of the introduction needs attention, organize the ideas, and ensure it flows logically and nicely for the reader. 

Authors' answer: Thank you, this is a valid point and we have added a section in the introduction on the consequences of unplanned pregnancies in low- and middle income countries. We chose to start introducing the RLP as this is the main subject of the study.

Methods: 

Setting: this section actually more relevant on the introduction section. The setting her should describe the “Research setting”, i.e. two low-income one large city and another rural area.

Authors' comment: Thank you, we have rewritten this section and the information about Eswatini and its particular reproductive health problems are now found in the introduction.

Study design section is missing.

Authors's comment: We have added information on page 7, specific information is presented below each subheading.

Study population: Who is the population of the study to which the results will be generalized? Here describing the sample rather than the population. Maybe start by describing the population and then followed by the sampling.

Authors' comment: Thank you for your comment, the population of the study is the Mentor Mothers that are a kind of community health workers. 

‘Fifty-three MMs were employed by Siphilile and 29 were asked to participate, as this number would include several MMs from each geographical region where Siphilile is active. We estimated three to four focus groups would be sufficient per data collection time point. All MMs agreed to participate; 20 MMs were working in the Manzini region and nine MMs in the Lubombo region.’

Are the MM considered in this study as a research team or as a study subjects/participants. They are reported to have been involved in training, they participated in FGDs and they also answered questionnaires, etc. their roles needs to be clarified, this will inform the writing of other sections i.e. if the MM are researchers then what about inter-rater validity and reliability. If they are participants had they been given the choice to participate and signed consents and were free to enroll or withdraw, etc. 

Authors' answer: The MMs are considered participants. This is described under the subheading ‘Setting and study population’.

Ethical concerns: Now this section has answered some of the queries raised before about the settings of the study. But lack main information about the voluntary participation of the MM in the study and how that was ensured, the confidentiality of the participants especially that the FGD was recorded and was done in their workplaces, etc. 

Authors' answer: Thank you, we have expanded this section to make it clearer and hope you find it sufficient.

The results of both the qualitative and quantitative components were presented together which added difficulty in reading the manuscript.

Authors' answer: Thank you for your comment. The qualitative and the quantitative was purposively presented together as we wanted to present the findings below each theme, as they are validating each other. However, we have added a clarifying statement in the first results section:

‘Qualitative as well as quantitative findings from Q2 are presented below sub-headings representing each of the constructs in the i-PARIHS framework (Fig 2) and the complete findings from Q2 are presented in Table 2. Quotes from FGDs held in January are labelled ‘FGD1’ and FGDs held in May are labelled ‘FGD2’, followed by the group number and the number of the MM who expressed it.’

REVIEWER 2

Overall this has been an interesting manuscript to read, and I suggest a useful tool to be used in the field for disadvantaged communities. The paper does however need some rework to make it suitable for publication.

Authors' comment: Thank you for your appreciation of our manuscript and for valuable feedback.

Firstly, in the Methodology Section, you need to outline in greater detail what the i-PARIHS Framework is. This would greater highlight its criticality to the study and provide the reader with some context.

Authors' comment: This is a good idea and we have added more information on the underlying theory of the i-PARIHS in the method’s section. We hope you find it sufficient.

Secondly, you need to be more explicit about the theme generation and how they were derived. This wasn't completely clear in the manuscript.

Authors' comment: Good point, we have expanded on this section: ‘Codes were grouped into the i-PARIHS constructs (facilitation, innovation, recipients and context), constantly going back to the transcripts to make sure our interpretations of the codes were valid. The same code could appear in more than one construct and codes that did not fit into any of the constructs were saved for potential later categorization.

We used thematic analysis by Braun et al. (1) and mainly applied a realist perspective as we sought to report the experiences and reality of the participants. After grouping the codes, they were printed on paper and we searched for patterns among the codes within each construct. 

After grouping the codes, they were printed on paper and we searched for patterns among the codes within each construct. We manually grouped codes that had a pattern capturing something relevant to the research questions and created themes based on these codes. Each theme was controlled to make sure that it was supported by the codes and if it was congruent with the entire data set. The creation of themes was undertaken in groups of two to three members of the research team during five work-shops, and different constellations of researchers participated in each work-shop. 

Finally, the themes were reviewed, refined and negotiated within the whole research team. The participants did not provide feedback on the themes.’

Thirdly, and most importantly, the Results section needs to be re-written. There are a number of long participant quotes that don't help explain the themes. The results section overall needs to be revised to show much stronger relationships and linkages to the themes outlined in the section. You have also included some statements in the results section that need participant quotes to support your argument (for example lines 413-414)

Authors' comment: Thank you, we agree with this critique and have now shortened several themes as well as added quotes to statements that were not supported before. We hope you now find it adequate.

REVIEWER 3

A brief description of the relationship between the PI and the Siphilile Maternal and Child Health (a non-governmental organization was provided. However, instead of only providing information about the PI’s familiarity with the management of Siphilile, some description of the PI’s familiarity with the local people including MMs might give the readers an idea on the trust is gained.

Authors' comment: Thank you for reviewing our manuscript and for giving important feedback. We have added more information on our relationship with the MMs and familiarity with the context under the subheading ‘Methodological considerations’. We hope you now find it sufficient.

Mentor Mothers (MMs) as sample leads to the question of whether it fits the idea of “train the trainer.” Some justification needs to be provided on why the RLP intervention was implemented among the MMs, and why not using the MMs as the trained community health workers to educate their people. The MMs have a higher awareness of health and are already more successful than other community women. It poses the question on how the RLP can be implemented among non-MMs considering the different characteristics they have.

Authors' comment: Thank you for highlighting the importance of training the trainer. However, we are not sure we understood your comment correctly but we hope this clarification will make it clearer:

As the RLP is a health tool aimed to be used by health care professionals, we chose to train the Mentor mothers in using it. As you write, a MM is a kind of community health worker, but has an important attribute in being a peer as well. 

The MMs train their clients in pregnancy planning and preconception health, thus, we have trained the trainer (the MM).

We chose to train the MMs as we hypothesized the trustful relationship they have with their client would be beneficial. Training other community health workers that are not peer supporters in using the RLP is a good suggestion for another study (mentioned in the discussion).

The authors state that the study used the i-PARIHS framework which offered a multifaceted picture of implementation of this study. This statement is not accurate because utility of the framework seems to be limited in the theme identification. The authors stated that codes were grouped into the iPARIHS constructs. The actual implementation of the RLP, the authors mentioned the use of NoMAD instrument in modifying question #2. Is the process of having a tailored RLP intervention also guided by the underlying philosophies from i-PARIHS.

Authors' comment: Thank you for your comment. We will try to address your comment point-by-point below:

The i-PARIHS is an organizing or conceptual framework to help explain and predict why the implementation of an intervention is or is not successful. It can be used to structurally describe implementation processes. The framework is flexible to use and the founders of it suggest it to be used as we have used it: as an evaluative tool within implementation practice and implementation research (2). We agree that our wording ‘multifaceted’ was not accurate and have now rephrased it to: ‘Using the i-PARIHS framework offered a structured reporting of implementation of the RLP and our themes fitted well into its constructs.’

The intervention was not guided by the iPARIHS framework as this was only a tool used in evaluation of the process. The i-PARIHS does not describe or discuss specific implementation strategies, such as tailoring of interventions. 

The NoMAD is a questionnaire that is used to measure or evaluate how well implemented an intervention/innovation is. We used the NoMAD in questionnaire 2 and the questions included are presented in the supplements.

---

## [Decision Letter · Decision Letter 1]

4 Jul 2020

PONE-D-19-35394R1

Evaluating the Implementation of the Reproductive Life Plan in Disadvantaged Communities: a Mixed-Methods Study Using the i-PARIHS Framework

PLOS ONE

Dear Dr.Jenny Niemeyer Hultstrand,

Thank you for submitting your manuscript to PLOS ONE. After careful consideration, we feel that it has merit but does not fully meet PLOS ONE’s publication criteria as it currently stands. Therefore, we invite you to submit a revised version of the manuscript that addresses the points raised during the review process.

Please submit your revised manuscript by 5th August, 2020.  If you will need more time than this to complete your revisions, please reply to this message or contact the journal office at plosone@plos.org. Please include the following items when submitting your revised manuscript:

We look forward to receiving your revised manuscript.

Kind regards,

Sharon Mary Brownie

Academic Editor

PLOS ONE

Additional Editor Comments (if provided):

Reviewers have returned comments with some final areas for improvement. Please address the issues raised.

Reviewers' comments:

Reviewer's Responses to Questions

**Comments to the Author**

1. If the authors have adequately addressed your comments raised in a previous round of review and you feel that this manuscript is now acceptable for publication, you may indicate that here to bypass the “Comments to the Author” section, enter your conflict of interest statement in the “Confidential to Editor” section, and submit your "Accept" recommendation.

Reviewer #1: (No Response)

Reviewer #3: All comments have been addressed

2. Is the manuscript technically sound, and do the data support the conclusions?

Reviewer #1: (No Response)

Reviewer #3: Yes

3. Has the statistical analysis been performed appropriately and rigorously? 

Reviewer #1: (No Response)

Reviewer #3: Yes

4. Have the authors made all data underlying the findings in their manuscript fully available?

Reviewer #1: (No Response)

Reviewer #3: Yes

5. Is the manuscript presented in an intelligible fashion and written in standard English?

Reviewer #1: (No Response)

Reviewer #3: Yes

6. Review Comments to the Author

Reviewer #1: Dear editor,

Thanks for asking me to review this manuscript again. The authors have done really a good job in revising their manuscript. They addressed majority of the comments and the manuscript reads much better now. I really enjoyed reading it to the end. I think the international readers will benefit from reading this article. I have only minor comments that I would like to share

Methods section at the beginning is very important and may need to be supplemented with a little more [brief] details before detailed description comes later on. There is a figure that explains to the reader the intended plan for the study, but again this is introduced later.

In line 110 [starting from “We…”] there was an abrupt move from describing the setting and the population to describing the sample and recruitment. I suggest separating this into a new section called sample and recruitment and add more details on this. Did you recruited the whole population to participate in the study, or you have selected part? How was the selection? Etc.

For me the use of the questionnaire in this study has added very little value, the rich discussions that took place in the group interviews has covered in more depth the areas that was asked about in the questionnaire. I would remove this part, in my opinion it will make the description of the study methods and reporting of the results more straightforward and easier. But I will leave this decision to the authors.

In the limitation section the authors have described their roles etc, I don’t see this as necessary.

Reviewer #3: Overall, the revised paper has addressed the questions I had. The only comment I have is to include in the limitation section that the study did not record the the number of RLP implementations each MN attempted to deliver and actually delivered during the intervention period. These data may provide some insights on MN's qualitative feedback.

7. PLOS authors have the option to publish the peer review history of their article (what does this mean?). If published, this will include your full peer review and any attached files.

Reviewer #1: No

Reviewer #3: No

---

## [Author Response · Author response to Decision Letter 1]

10 Jul 2020

Reviewer #1: 

Dear editor, thanks for asking me to review this manuscript again. The authors have done really a good job in revising their manuscript. They addressed majority of the comments and the manuscript reads much better now. I really enjoyed reading it to the end. I think the international readers will benefit from reading this article. I have only minor comments that I would like to share

Authors’ response: Thank you for constructive comments and we are happy that you think the manuscript has improved. We have responded to the comments point-by-point below.

1. Methods section at the beginning is very important and may need to be supplemented with a little more [brief] details before detailed description comes later on. There is a figure that explains to the reader the intended plan for the study, but again this is introduced later.

Authors’ response: Thank you, we have added a few more details in the description and moved figure 1 ‘Study procedures and timeline’ to the first paragraph in the methods section (page 7, line 87-91). We agree this increases readability.

2. In line 110 [starting from “We…”] there was an abrupt move from describing the setting and the population to describing the sample and recruitment. I suggest separating this into a new section called sample and recruitment and add more details on this. Did you recruited the whole population to participate in the study, or you have selected part? How was the selection? Etc.

Authors’ response: Good suggestion, we have separated this information into a new section (page 8, line 112-119). 53 MMs were eligible and we estimated 29 informants would be sufficient. The selection of MMs was based on the geographical area where they are assigned to work, as we wanted to have representatives from each region. Discussions on data saturation were held continually and we chose to not include more MMs as we deemed data saturation was reached in the primary analysis of the results.

3. For me the use of the questionnaire in this study has added very little value, the rich discussions that took place in the group interviews has covered in more depth the areas that was asked about in the questionnaire. I would remove this part, in my opinion it will make the description of the study methods and reporting of the results more straightforward and easier. But I will leave this decision to the authors.

Authors’ response: We wish to keep the questionnaire as we think it adds credibility to the findings (as discussed on lines 562-566) and as the use of the NoMAD questionnaire has not been reported from any low- or middle-income setting before.

4. In the limitation section the authors have described their roles etc, I don’t see this as necessary.

Authors’ response: We wish to keep this information as it is required in the COREQ (Consolidated criteria for Reporting Qualitative research) Checklist.

Reviewer #3: 

Overall, the revised paper has addressed the questions I had. The only comment I have is to include in the limitation section that the study did not record the number of RLP implementations each MN attempted to deliver and actually delivered during the intervention period. These data may provide some insights on MN's qualitative feedback.

Authors’ response: Thank you for valuable feedback and we are happy that you feel we have addressed the questions you had. We monitored the number of RLP interventions by using report sheets, where the MMs filled out how many clients they had approached. We did not intend to use this information in the research mainly, rather as a way of facilitate implementation of the intervention, but we agree that this would have added valued when interpreting the results. There was a problem with lack of copies of the report sheets and of losing the report sheets. Therefore, one of the MMs suggested that this reporting would be included in the ordinary patient records and this have now been amended. As you suggest, we have added this as a limitation in the methods section (page 28, line 567-568).

---

## [Editor Report · Decision Letter 2]

14 Jul 2020

Evaluating the Implementation of the Reproductive Life Plan in Disadvantaged Communities: a Mixed-Methods Study Using the i-PARIHS Framework

PONE-D-19-35394R2

Dear Dr. Jenny Niemeyer Hultstrand,

We’re pleased to inform you that your manuscript has been judged scientifically suitable for publication and will be formally accepted for publication once it meets all outstanding technical requirements.

Kind regards,

Sharon Mary Brownie

Academic Editor

PLOS ONE

Editor Comments 

Reviewer decisions have been satisfactorily addressed.

---

## [Editor Report · Acceptance letter]

20 Aug 2020

PONE-D-19-35394R2 

Evaluating the Implementation of the Reproductive Life Plan in Disadvantaged Communities: a Mixed-Methods Study Using the i-PARIHS Framework 

Dear Dr. Niemeyer Hultstrand:

I'm pleased to inform you that your manuscript has been deemed suitable for publication in PLOS ONE. Congratulations! Your manuscript is now with our production department. 

Kind regards, 

on behalf of

Professor Sharon Mary Brownie 

Academic Editor

PLOS ONE